# CRISPR/Cas: A New Tool in the Research of Telomeres and Telomerase as Well as a Novel Form of Cancer Therapy

**DOI:** 10.3390/ijms23063002

**Published:** 2022-03-10

**Authors:** Mahendar Porika, Radhika Tippani, Gabriele Christine Saretzki

**Affiliations:** 1Department of Biotechnology, Kakatiya University, Warangal 506009, Telangana, India; prkmahendar@gmail.com (M.P.); tippanira@gmail.com (R.T.); 2Biosciences Institute, Campus for Ageing and Vitality, Newcastle University, Newcastle upon Tyne NE4 5PL, UK

**Keywords:** genome editing, CRISPR/Cas, telomeres, telomerase, aging, cancer, therapy

## Abstract

Due to their close connection with senescence, aging, and disease, telomeres and telomerase provide a unique and vital research route for boosting longevity and health span. Despite significant advances during the last three decades, earlier studies into these two biological players were impeded by the difficulty of achieving real-time changes inside living cells. As a result of the clustered regularly interspaced short palindromic repeats (CRISPR)-associated system’s (Cas) method, targeted genetic studies are now underway to change telomerase, the genes that govern it as well as telomeres. This review will discuss studies that have utilized CRISPR-related technologies to target and modify genes relevant to telomeres and telomerase as well as to develop targeted anti-cancer therapies. These studies greatly improve our knowledge and understanding of cellular and molecular mechanisms that underlie cancer development and aging.

## 1. Introduction

The previous century has seen an incredible increase in human life expectancy across the world. However, this rise in life span has not been supported by a similar increase in health span [1,2]. Aging is the leading cause for most pathological conditions such as immunosenescence, cardiometabolic disorders, osteoporosis, sarcopenia, arthritis, cataracts, neurological diseases, and most malignancies that shorten life expectancy and increase chronic illnesses in older people [3].

Telomeres are highly specialized nucleoprotein complexes that play a critical role in cell senescence and aging. Each chromosomal end must be “capped” with a critical amount of telomere repeats to prevent DNA repair pathways from being activated [4,5]. Telomeres consist of a six-protein complex called shelterin that binds to telomeric DNA containing tandemly repeated hexanucleotides (TTAGGG in mammals, including humans). Human telomeric DNA is about 5 to 15 kb in length and is composed of double-stranded (ds) telomeric repeats that end in a single-stranded (ss) 3′ G-rich overhang [6,7]. In addition, telomeres and shelterin form a telomeric T-loop and a displacement D- loop [8]. The latter sequesters the ss overhang away and protects it from degradation and emanation of a DNA damage signal. With each DNA replication, telomeric DNA shortens by around 50–200 bp due to the end replication problem (ERP) of semiconservative DNA replication where DNA polymerase can synthesize only the leading strand continuously while the lagging strand is synthesized with the help of an RNA primer and short DNA fragments (Okazaki fragments). Those are finally stitched together by DNA ligase while the most distal RNA primer cannot be replaced by new DNA and its removal thus leaves an ss overhang of around 100–200 nucleotides [9]. In addition, oxidative stress can accelerate telomere shortening [10]. Telomeres shorten to the point of a minimum length when they cause a DNA damage response (DDR) that can lead to cellular senescence or apoptosis [4,11,12]. In the absence of additional modifications, cells can survive and remain senescent for years, and that is regarded as a tumor suppressor mechanism in long-lived species such as humans. Senescent cells release various types of molecules and have a senescence-associated secretory phenotype (SASP) that can target neighbouring cells and promote age-related disorders [13,14]. As a result, with progressing age, there is a steady build-up of senescent cells in various tissues. In contrast, removing those senescent cells from the organism either by genetic means or the use of senolytics is able to delay and ameliorate many age-related conditions in animal models [15,16].

The enzyme telomerase is able to counteract telomere shortening by adding telomeric repeats *de novo* to telomeres. It thereby increases the proliferative capacity of cells with high amounts of telomerase such as embryonic stem cells and cancer cells [17]. Telomerase is a RNA-dependent DNA polymerase and reverse transcriptase consisting of the reverse transcriptase protein (TERT) and a RNA component (TERC) that contains the template region for telomere synthesis. Telomeres gradually shorten in the absence of telomerase activity (TA) due to the ERP. In most human somatic cells there is no TA because *TERT* is not expressed while TERC is generally present. TA may be reintroduced into human cells using *hTERT* (human TERT) gene transduction resulting in an extended lifespan due to continuous telomere maintenance preventing senescence and crisis. However, for a cell to be immortalised and tumorigenic, additional molecular events are required such as genetic instability, oncogene activation and tumour suppressor inactivation.

TA is suppressed in most tissues very early during human development [18] while TA is present in germ-line cells, embryonic and adult stem cells, and B- and T-lymphocytes as well as endothelial cells [19]. In contrast, TA is reactivated in the majority of human cancer cells, in many cases due to *hTERT* promoter mutations [20,21], suggesting a possible target for cancer diagnosis and treatment [22,23,24,25,26,27]. As a result, various telomerase antagonists have been created to combat cancer; unfortunately, none has become a suitable clinical treatment option due to adverse side effects [25,27]. In addition to the main function of telomerase in telomere maintenance the protein subunit TERT has various non-telomeric functions that can contribute to tumorigenesis [28] and play a major role in neurons and the brain [29].

## 2. The CRISPR/Cas System 

Despite tremendous advances in the understanding of functions and dysfunctions of telomeres and telomerase, the inability to influence and detect changes directly inside biological systems has been an essential hurdle in their investigation. However, a solution to this matter has recently been developed—the clustered regularly interspaced short palindromic repeats (CRISPR)-associated system (Cas). For the first time it was reported in the early 2000s as an adaptive bacterial immune system that targets and destroys invading bacterial and viral DNA in prokaryotes [30]. It has since gained widespread acceptance. This finding generated evolutionary curiosity when it was identified because of its genomic CRISPR and Cas protein. However, it took an additional eight years to find and acknowledge an application of this system. Different groups presented ways of editing DNA in vitro using the CRISPR/Cas system in 2013 and 2014 [31,32,33]. 

This method is composed of two fundamental categories—guiding and effecting. An ssRNA molecule known as single guide RNA (sgRNA) serves as the guiding portion responsible for specificity [34]. This sgRNA component addresses a genomic area by complementing a specific DNA sequence and is linked to numerous Cas proteins, the most prevalent of which is Cas9. This protein exhibits double-strand (ds) endonuclease activity in its natural state [32,35] (Figure 1). Donor DNA with the required sequence can be introduced into the target area once a cut in the DNA has been created [36]. By pairing endonuclease activity with the RNA guide, genetic information can be changed in vitro and in vivo in a highly precise and carefully controlled manner. 

However, the CRISPR-Cas system is not confined to only causing double-strand breaks (DSB). It is possible to modify the Cas protein to preserve its targeting capability while it loses its ability to cut DNA [34]. Cas/dead Cas (dCas) (the latter is catalytically inactive) can be used as it is or be modified with various functional groups. Molecules (labels and effectors) can be attached to dCas as shown in Figure 1. These molecules can then be placed closer together or be associated to specific areas of the genome. 

Through innovation CRISPR has been developed into a powerful bio-analytical method to detect nucleic acids and diagnose various diseases [38,39,40,41,42]. A short CRISPR RNA directs CRISPR/Cas12a to select a target dsDNA and generates the Cas12a/crRNA/dsDNA ternary complex which cleaves the target dsDNA (cis-cleavage). Cas12a’s capacity to trans-cleave random ssDNA is triggered by the ternary complex formation following target DNA cleavage [38,43,44]. CRISPR therapies, which were discovered using CRISPR, are being developed into a potential therapy for malignancies and other genetic abnormalities [45].

RNA-guided immune mechanisms, such as CRISPR, are used by bacteria and archaea to fight off invaders such as viruses and plasmids [46]. In the presence of an adjacent protospacer motif (PAM) on the opposite strand, the sgRNA is able to direct CRISPR/Cas9 to a target location for cleavage [47], leading to DNA DSBs (Figure 2) [31,48]. The sequence-specific dsDNA binding and cleavage capabilities of the Cas9/sgRNA or dCas9/sgRNA systems have been used to develop biosensors for nucleic acid assays [40,47,49]. Because of characteristics such as adaptability, simplicity, specificity, and effectiveness, CRISPR/Cas9 technology has been extensively employed for genome editing (Figure 3) and has a significant potential for biomedical studies [50,51].

## 3. Targeting Telomeres with CRISPR

The ability to apply CRISPR imaging directly to living biological systems is the real value of the technique. Other fluorescently labelling genomic approaches exist, however they are harmful to cells and may result in irreparable DNA damage [52]. In vitro recording of telomeres and other genomic components has been unachievable due to this restriction. Targeting telomeres for imaging is a new application of the CRISPR/dCas system. This method offers several benefits over previous systems with the majority of benefits being due to the CRISPR system’s dynamic and long-term persisting nature. Among the first imaging investigations performed with CRISPR was one performed by Chen and colleagues [53]. They used enhanced green fluorescent protein (EGFP) to detect telomeres in HEK293T, UMUC3, and HeLa cell lines. The authors identified telomere movements within these cells with a labelling efficacy and intensity similar to the well-known DNA FISH procedure.

Further optimization to this method may result in further improved labelling efficiency and specificity [54]. Labelled telomeres became much simpler to identify when the EGFP fluorescent tag was replaced with the brighter mClover fluorescent tag, which however, resulted in minor off-target impacts [55]. Although imaging of telomeres is not a unique concept, the CRISPR/Cas system’s exceptional accuracy and efficiency form a novel technique to track telomeres fast and effectively.

Shao and co-workers were the first to demonstrate that CRISPR-dCas labelling was minimally cytotoxic to cells and appropriate for long-term observations [56]. Their technique was utilized to quantify the relative movements of telomeres and centromeres during interphase within a five-minute timeframe [56]. This approach has recently been applied to transgenic mouse models [57]. By expressing dCas-GFP throughout a mouse, telomere guides might be inserted into particular tissues for labelling. The authors combined this method with CRISPR-interference of the *TRF1* gene to detect telomere aggregation and fusion in real-time [57]. This approach can be applied to additional genes, allowing investigators to analyze changes in telomere dynamics following genetic modifications. 

Inducible shelterin CRISPR/Cas9 knockout (KO) cells were used in a study conducted by Kim and colleagues to perform a thorough investigation of human telomeric dysfunction [58]. The authors developed inducible CRISPR KO human cell lines for the shelterin complex subunits TRF1, TRF2, RAP1, TIN2, and TPP1, as well as POT1. In mice, homozygous ablation of many of those subunits causes death [58]. The majority of human telomere regulator loss-of-function investigations utilized RNA.

Interference-mediated gene knockdown, has its own set of limitations. Thanks to the inducible CRISPR technique, the authors collected huge numbers of KO cells with critical telomere regulators deactivated for a rapid biochemical and molecular downstream investigation. This study discovered functional differences between human and mouse telomeric proteins in DDR, telomere length regulation, and metabolic control [58]. 

Dai and co-authors created a CRISPR/Cas9 protein in tumor cells through an NF-κB-activated gene-expression (Nage) vector [59]. Due to sgRNA targeting telomeric DNA co-expressed in cells, Cas was able to cleave telomeric DNA, which resulted in tumor cell death. An adeno-associated virus (AAV) which contains the Cas9/sgRNA expression vector can be packaged and delivered intravenously into mice in order to inhibit tumor growth without generating side effects or toxicity [59].

DDR signaling initiates and maintains cellular senescence [11]. DDR signaling, genomic instability, and cellular senescence can all be triggered by telomere dysfunction, and the connections between these processes have been well-studied. Abdisalaam and co-workers employed a combination of biochemical and imaging approaches to induce DNA DSB selectively in telomeres, along with a highly regulatable CRISPR/Cas9 approach [60]. Moreover, the authors used micronucleus imaging, telomere immunofluorescence, fluorescence in situ hybridization (FISH), chromatin immuno-precipitation (ChIP), and the telomere shortest-length assay (TeSLA). They demonstrated that chromosomal mis-segregation caused by induction of DDR signaling in response to faulty telomeres resulted in cytosolic chromatin fragments, leading to an early senescence phenotype [60]. The authors found that cytosolic chromatin fragments were recognized by cyclic GMP–AMP synthase (cGAS), which activated the stimulator of interferon genes (STING) cytosolic DNA-sensing pathway and downstream interferon signaling. Not only did genetic and pharmacological alteration of cGAS diminish immunological signaling, but it also reduced telomere dysfunction-related premature cellular senescence [60].

An improved approach in CHO (Chinese hamster ovary) and mouse A9 cells following microcell-mediated chromosome transfer (MMCT) to recipient cells utilizing a CRISPR/Cas9-induced homologous recombination (HR) was demonstrated by Uno et al. [61]. CRISPR/Cas9 and a circular targeting vector comprising two 3 kb HR arms were used to introduce EGFP into CHO cells. CRISPR/Cas9 and a linearized truncation vector with a single 7 kb HR arm at the 5′ end and a 1 kb artificial telomere at the 3′ end were used in CHO cells to accomplish telomere-associated truncation [61] For transgene insertion and telomere-related truncation, 6–11% of the targeting efficiency was obtained. A9 cells were used to confirm the transgene insertion (29%) [61]. The modified chromosomes can be transferred to other cells. As a result, chromosomal engineering using CHO and A9 cells as well as CRISPR/Cas9 for direct chromosome transfer is a rapid technique that will enable easier chromosome modification.

In the neuroblastoma cell line SH-SY5Y Kim and colleagues used the CRISPR-Cas9 tool to remove telomeres and promote senescence [62] Expression of Cas9 and guide RNA targeting telomere repeats ablated telomeres, resulting in reduced cell growth. Telomere deletion also impacted mitochondrial function in SH-SY5Y cells, resulting in lower mitochondrial respiration and cell survival [62]. Changes in the levels of Parkinson’s disease (PD)-associated proteins, such as PTEN-induced putative kinase 1 (PINK1), parkin, peroxisome proliferator-activated receptor coactivator- 1 alpha (PGC-1α), nuclear respiratory factor 1 (NRF1), and aminoacyl tRNA synthetase complex interacting multifunctional protein 2 (MFP-2), were demonstrated, lending support to the pathological relevance of cell senescence. Importantly, α-synuclein expression in the context of telomere elimination increased protein aggregation, suggesting a positive feedback loop between senescence/aging and PD pathogenesis [62]. The ablation of telomeres resulted in cellular alterations, including the loss of mitochondrial function and the accumulation of PD-related proteins as well as the reduction of cell survival. Because of these modifications, aging and PD-related features could be more easily replicated in cells. 

Telomere disruption and destruction, as described above, can result in a variety of cellular abnormalities. The capacity of the CRISPR-Cas system to cut and insert genes enables in vivo studies of telomere deterioration in real-time. The activation of a telomeric repair mechanism mediated by the Rad51 gene was triggered by this technique to produce DSBs in telomeres [63]. 

Several studies investigated the relationship between telomeres and senescence [11,62,64,65]. It is also possible to employ CRISPR/Cas to generate more moderate changes in telomeres. Cells lost sister telomeres and had a lower replication capacity when a mutation to a subtelomeric CTCF (CCCTC-binding factor) binding region that is involved in telomere transcription generating TERRA (telomere repeat-encoding RNA) was introduced [66]. The creation of replication stress with thymidine or aphidicolin worsened this problem, resulting in a higher amount of apoptosis [66]. It has been shown that CTCF and TERRA sites are critical for proper replication and maintenance of telomeres and the overall integrity of chromosomes. 

While the number of current studies on CRISPR/Cas-induced telomere deletion is limited, the technique’s ability to be applied to any cell type makes it possible to study a broad spectrum of disorders. In addition, the effect of senescence may be assessed across a wide range of settings in different tissues. Telomere shortening and its induced disorders (telomeropathies) may display new biological and mechanistic aspects that could help to better understand and characterize them.

## 4. Targeting Telomerase by CRISPR/Cas

While imaging the *TERT*-containing genomic area using dCas is conceivable, the nucleotide sequence is not the most critical factor for biological activity. As a result, it appears that altering and targeting the TERT protein is the most efficient tool to image and investigate its dynamics. Schmidt and colleagues detected three phases of TERT migration by inserting a fluorescent marker into the *TERT* gene [67]. An early diffusion stage and an intermediate telomere association stage, as well as the final long-term association stage were characterized as the three distinct phases in the progression of telomere elongation. Telomerase connects to the telomeric 3e ssDNA overhang and adds several TTAGGG hexanucleotides in a coordinated manner [68]. These findings allow a unique telomere extension paradigm to be built, in which telomeres are elongated in brief regulated intervals after prolonged periods of temporary connection of telomerase to the telomere. The capacity to analyze the spatiotemporal dynamics of telomerase migration and staffing to telomeres is unparalleled because of the CRISPR/Cas system’s labelling and subsequent direct imaging of telomerase. Addressing this dynamics is critical in the research of diseases that involve the enzyme telomerase to allow fast, unregulated cell division, like cancer. 

The most extensively used approach for determining telomerase activity in cells and tissues is the telomeric repeat amplification protocol (TRAP). Unfortunately, inhibiting substances in samples of body fluids or tumor tissue can suppress the TRAP reaction, resulting in false-negative findings. In a further advance of the technique, Cheng and co-workers created a colorimetric coding system based on a programmable CRISPR/Cas12a system and a gold nanoparticle (AuNP) probe to analyze telomeric repeat DNA and an internal control, enabling the naked eye to detect TA [69]. The authors converted the outcomes of the finding into positive, negative, and false-negative outcomes, making it simpler and more precise to evaluate detection results. The technology was also utilized to reliably identify TA in liver cancer samples, with 93.75% detection sensitivity and 93.75% specificity, according to the Youden index analysis [70]. As a proof-of-concept, the authors developed a Cas9-mediated triple-line lateral flow assay (TL-LFA), allowing them to detect telomeric repeat DNA and an internal control on a single triple-line test strip, resulting in a practical and efficient TA assay [69].

Yu and colleagues created an all-in-one TA test using the telomere-synthesis-induced CRISPR-Cas12a system [71]. Extension of the telomere by telomerase resulted in telomere repeats (TTAGGG)n, which could be recognized using a customized CRISPR-guided RNA (crRNA) and activated by the conventional trans-cleavage CRISPR- Cas12a detection assay [72]. Because CRISPR-Cas12a is so sensitive, this method achieved a linear regression spanning 100 to 2000 HeLa cells and a detection limit of only 26 HeLa cells [71]. Furthermore, the suggested approach allows the detection of TA in a single tube under isothermal conditions (37 °C) utilizing a simple and rapid procedure. This all-in-one technique has a lot of potential in point-of-care (POC) TA detection because of its high performance.

Using CRISPR-Cas12a and rolling circle amplification (RCA) (method of isothermal amplification of circular DNA molecules) (Figure 4), the team of Zhou et al., developed an in vitro method for the detection of TA [73]. Without adding dATP nucleotides, primers for telomerase substrates (TS) can be simply extended by five bases (GGGTT) to provide short telomerase extension products (TEP) with specific lengths. The generated short TEPs can then cyclize the padlock by hybridizing with its two terminals, triggering the following subsequent RCA. After detecting the target ssDNA sequence in the RCA products, the CRISPR-Cas12a system is coupled to collaterally cut surrounding DNA reporter probes to increase sensitivity [73]. The key points of the method are: (i) detection of low TA levels is improved with the use of a brief TEP-triggered method, hence aiding in the early detection of human cancers; (ii) RCA-initiated CRISPR/Cas12a provides very sensitive TA detection that is simple to use; and (iii) the technique offers a new path for TA detection using a CRISPR-Cas12a system with considerable potential application for the diagnosis of malignancies. 

Wei and colleagues created a unique approach for detecting simultaneous TA and TERT levels in osteosarcomas using an ingeniously designed stem-loop probe and CRISPR-Cas12a [74]. The authors demonstrated that the suggested method is able to effectively detect cancer cells equivalent to traditional TRAP assays in both laboratory and clinical studies [74]. In addition, they discovered that average TA might be used to grade osteosarcomas. Consequently, the technique would open up a new dimension for bioanalysis and disease diagnostics for osteosarcomas by combining three indicators in one detection, including TA and TERT levels.

Knocking in or out a gene in order to change gene expression is a common technique and an important method in biotechnology. When comparing differences of mutants to wild-type (WT) organisms, the genetic function of the gene of interest can be evaluated. The CRISPR/Cas technique can both ablate and boost *TERT* expression by focusing on the *TERT* promoter [75]. Mutations that result in *TERT* silencing usually cause immortal cell lines to senescence or die while those that enhanced expression exhibited TERT levels comparable to tumor cell lines [75]. Chiba and co-authors discovered that by attaching a protein Halo-tag to the N-terminus of the TERT protein they could modulate TA between these two described extremes [76]. As expected, the authors found a decrease in telomere length in cell lines when they inhibited TERT. 

Similar associations were investigated by Xi and co-workers in a separate study focusing on urothelial carcinoma cells [77]. In the *hTERT* promoter of these cells, there is a single DNA substitution mutation that had been previously linked to high *TERT* expression levels. The authors used the CRISPR/Cas system to reverse the mutation which resulted in restoration of *hTERT* levels to baseline levels of the wild-type sequence. Because *TERT* promoter mutations are so common in cancer cells, it is crucial to understand their significance for cell growth and tumorigenicity [78]. The mechanism behind this phenomenon was investigated in a subsequent study. The *TERT* gene promoter mutations were induced by using the CRISPR-Cas technology [79]. Chromatin interaction upstream of the gene increased, and recruitment of the transcription factor GABPA (GA binding protein *transcription factor* subunit alpha) was observed in these mutants [79]. This transcription factor directly recruits DNA polymerase II, and this might be a mechanism for *TERT* activation in a variety of malignancies caused by promoter mutations. 

While directly targeting telomerase in cancer therapy, unwanted side effects of telomere lengthening without a matching rise in TA can still occur occasionally in some cancers [80]. The tumors can still grow without TA because they maintain telomeres and avoid senescence with an alternative lengthening of telomeres (ALT) mechanism which is based on homologous recombination of telomeres [81,82]. Creating cell lines devoid of TA is a way to gain a better insight into the ALT pathway and cancer cells using this telomere-maintaining mechanism.

Cells with an ALT pathway were created experimentally by using CRISPR to knock out *TERT* and the ALT-related pathway *ATRX/DAXX* (alpha thalassemia/mental retardation syndrome X-linked chromatin remodeler/death domain-associated protein), which is often lost in ALT-positive cell lines while the mechanism is not entirely clear [83]. This ALT route may also be accomplished by CRISPR-mediated knockout of the telomerase RNA component (TERC) [84]. However, with this technique only a small percentage of cells developed the ALT phenotype. In the cells which developed ALT, it resulted in telomeres with extensive overhangs on the lagging strands [84]. While telomere elongation with ALT appears to be relatively uncommon, a better understanding of the mechanism and biology of the process is crucial for investigating cancers that do not use telomerase-associated telomere lengthening. 

CRISPR/Cas is a robust technique for introducing mutations into the DNA of living cells and modelling diseases in genetically engineered cell lines. This modelling, however, is not confined to cancer cell studies. It offers a lot of potential in modelling aging and age-related diseases caused by cellular senescence. For example, using the CRISPR-Cas method a study generated a RecQ like helicase (WRN) gene KO in human embryonic stem (ES) cells resulting in a new Werner’s disease model [85]. 

With an improved understanding of disease pathways, the ability to generate appropriate cellular models for many diseases has never been more crucial. These models can be employed to evaluate novel types of drugs and treatments. Thus, it is critical to develop and use disease specific models in order to guarantee that therapies work in individuals, an approach also known as “personalized medicine”. 

A telomerase-activated gene-expression (Tage) system was established by Dai et al. [86] in order to analyze TA in malignant cells. The Tage system included an effector gene expression vector with a telomerase-recognizable 3′ sticky end and an artificial transcription factor expression vector that expresses *dCas9-VP64* and a sgRNA targeting the telomere repeat sequence. The Tage system was capable of killing numerous tumor cells (HepG2, HeLa, PANC-1, MDA-MB-453, HT-29, and SKOV-3) by employing Cas9, while not affecting genetically normal cells such as MRC-5, HL7702, or bone marrow mesenchymal stem cells (BMSC) [86]. This discovery was critical because it allowed the Tage system to efficiently kill cancerous cells in vivo using telomerase to identify a four-base 3′-sticky end generated by homothallic switching endonuclease (a sequence-specific endonuclease from yeast) [86]. The Tage system achieved its desired results in vivo by using AAV as a vector. The Tage system is capable of selectively destroying tumor cells in mice without causing any adverse effects or harm [87] (Figure 4). 

Inactivating *TERT* has been identified as a viable cancer therapy by Wen and colleagues [87]. The authors used the CRISPR/Cas9 gene-editing method to target the *TERT* gene in tumor cells, finding that disrupting *TERT* affects tumor cell viability both in vitro and in vivo. In vitro, telomere attrition and growth retardation can be caused by *TERT* haploinsufficiency in tumor cells [87]. *TERT*-haploinsufficient tumor cells failed to produce xenografts in nude mice following transplantation, demonstrating that gene editing-mediated *TERT* deletion is a promising cancer treatment approach [87].

A CRISPR/dCas9-guided and telomerase-responsive nanosystem has been created by Ma and colleagues for nuclear targeting and adaptive release of antitumor drugs [88]. Doxorubicin (DOX), an effective chemotherapeutic agent, was conjugated to mesoporous silica nanoparticles (MSNs) using the CRISPR/dCas9 technique. To enclose DOX into MSNs, a specifically designed wrapping DNA was employed as a telomerase-responsive biogate (MSNs/DOX/DNA). The wrapping DNA is extended in the presence of telomerase, which is highly active in cancerous cells but not in normal ones. The extended DNA sequence diffuses far from the MSN surface, forming a stiff hairpin-like shape [88]. Targeting telomere-repeat regions using CRISPR-dCas9 ensures a proper delivery of the nanosystem to the nucleus and an efficient drug release mediated by telomerase. This enables precise drug release. An anti-cancer drug delivery method and nanosystem were proposed in this study, and resulted in enhanced tumor cell killing efficiency by nuclear-targeted delivery and tumor-specific drug release [88].

CRISPR/Cas13d was engineered for detection/sensing of hTERT utilizing the innovative device OFF-switch aptazyme (synthetic molecules composed of an aptamer domain and a catalytically active nucleic acid unit) and inserted into the Cas13d’s 3′ UTR by Zhuang and co-authors [89]. The Cas13d degradation in bladder cancer cells was prevented by the hTERT ligand coupled to the aptamer in the OFF-switch hTERT aptazyme. Results showed that modified CRISPR/Cas13d sensing hTERT lowered proliferation and migration of bladder cancer cells in vitro and induced cell death without affecting normal human foreskin fibroblasts, indicating that it might be utilized to specifically treat bladder cancer cells [89]. 

## 5. Conclusions 

Aging is a complex and continuous process defined as an organism’s constant decline in physiological functions. On the cellular level, telomeres shorten in dividing cells and get damaged in post-mitotic cells resulting in senescence. Only a very small number of human cells (adult stem cells, lymphocytes, endothelial cells, and germline cells) possesses telomerase activity (TA), which is able to counteract telomere shortening by adding new telomere sequences. In contrast, TA is present at a constitutively high level in cancer cells, which supplies those cells with an unlimited proliferation potential. CRISPR/Cas technology, which was recently developed, has been harnessed as a powerful tool for generating quick and precise genetic modifications in biological systems. As outlined in this review, this approach has already contributed much to our understanding of telomeres and telomerase in the context of cancer, senescence, and aging, and it will surely continue to do so as the technology advances further. Importantly, the CRISPR/Cas technology has a potential to kill cancer cells and thus to be used as a novel, promising anti-cancer therapy without the so-far-known side-effects of other strategies.

## Figures and Tables

**Figure 1 ijms-23-03002-f001:**
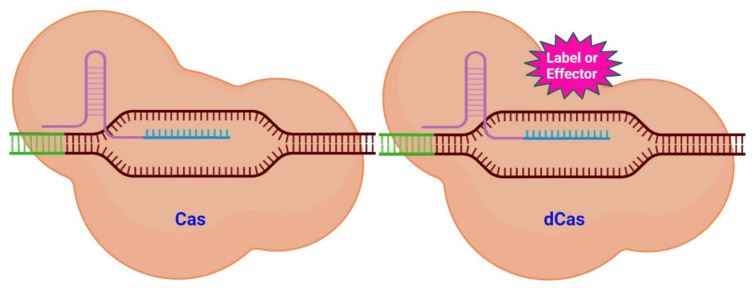
Scheme of CRISPR/Cas system variants. Targeting a specific genomic region (brown) with its sgRNA (lilac and turquois) is the function of the Cas system (**left**). The Cas protein will subsequently cut into the neighboring DNA region (green). The dCas system (**right**) uses a similar mechanism to that of the CAS system to target a genomic region. The dCas protein, however, lacks endonuclease activity. Numerous compounds may be fused to the dCas protein (pink) (labels and effectors). Labels place a fluorescent signal near to the target DNA, while effectors are able to alter the epigenetic state of the DNA. Modified from [37] and generated with BioRender software.

**Figure 2 ijms-23-03002-f002:**
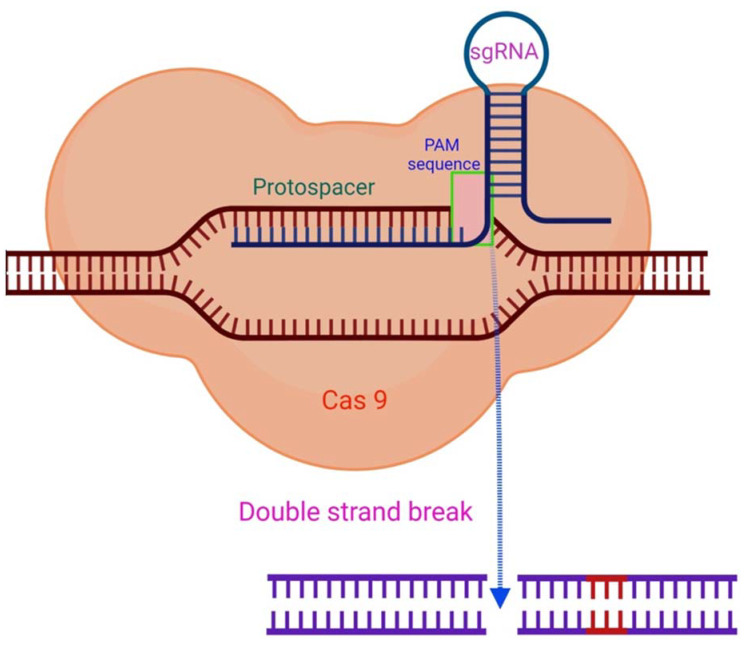
The CRISPR/Cas9 mechanism. By complementary base pairing, the sgRNA recognizes the target sequence (protospacer) in a host organism’s genome. The Cas9 nuclease causes a DSB in a PAM sequence region with a NGG (any nucleobase followed by two guanine nucleobase) sequence at the 3′position. Adapted from [47] and generated with BioRender software.

**Figure 3 ijms-23-03002-f003:**
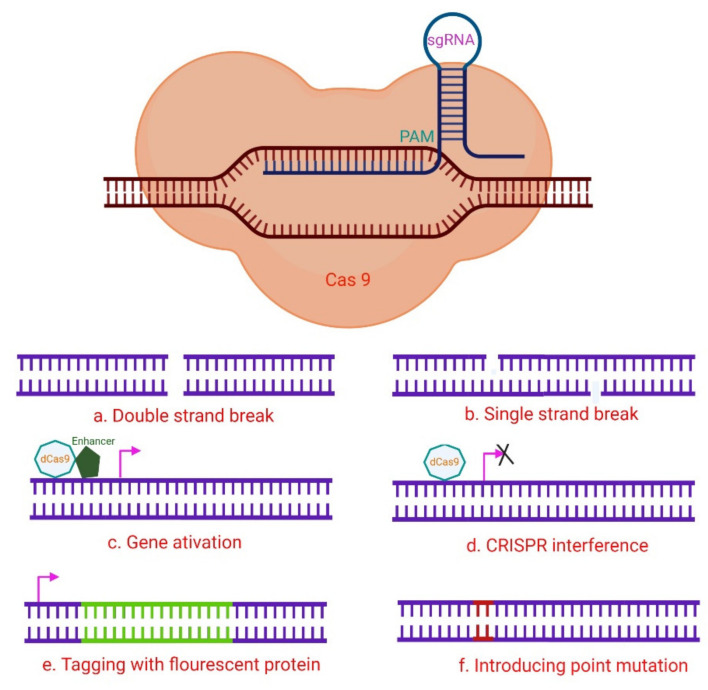
A hypothetical CRISPR/Cas9 experimental design. The numerous Cas9 variants available have made it easier to perform a variety of genomic modifications, including (**a**) induction of DNA DSB utilizing wild-type Cas9; (**b**) Cas9 nickase is used to induce DNA SSBs, in which two neighboring gRNAs target distinct strands, resulting in a DSB and subsequent NHEJ (non-homologous end joining) response; (**c**) The target gene is activated via CRISPR activation utilizing dCas9 and transcription activator fusion with transcription-activating proteins such as VP64 that target the promoter region. (**d**) By steric hindrance, CRISPR interference with dCas9 or dCas9-BFP decreases gene expression, and the dCas9-KRAB fusion epigenetically marks the gene for transcriptional repression. These methods can be used to stop gene transcription transiently. (**e**) A donor template that includes a fluorescent protein gene with homology regions can be used to drive homology-directed repair (HDR). By cutting at the target location with Cas9-WT or Cas9 nickase, it is possible to tag genes with fluorescent proteins. In another setting, the donor plasmid might include a selectable marker for determining whether knock-in events do occur. (**f**) Cas9-WT or Cas9 nickase also initiate the introduction of point mutations via HDR from an ss oligonucleotide. Adapted from [47] and generated using BioRender software.

**Figure 4 ijms-23-03002-f004:**
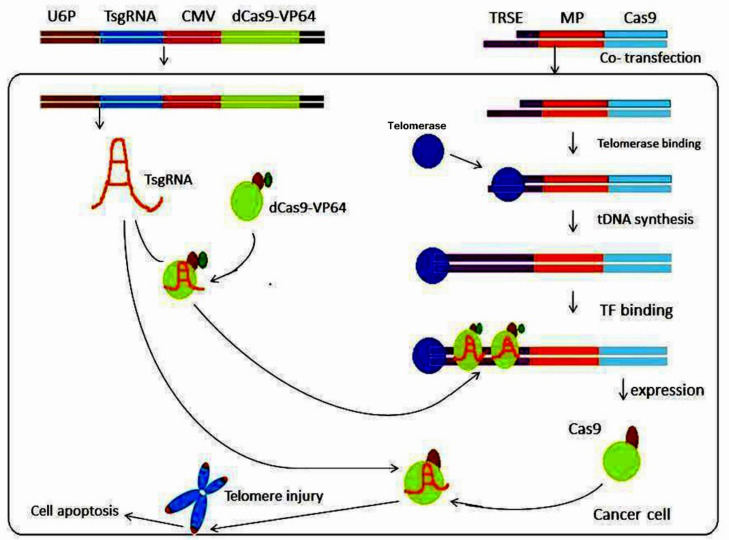
Schematic depiction of the Tage system’s cancer-killing mechanism. In this technique, cancer cells are transfected with an effector containing a telomerase-recognizable 3′ ss sequence that can be extended by telomerase and yield a synthesized ds telomeric repeat sequence. The dCas9-VP64-TsgRNA complex, which recognizes and binds to telomerase-synthesized ds telomeric repeat sequences, is generated by co-transfecting cancer cells with a sgRNA targeting telomeric DNA (TsgRNA) and an artificial transcription factor expression vector (dCas9-VP64). In this way, an effector gene known as *cas9* can be triggered. Cancer cells can be killed by the Cas9-TsgRNA complex, which can cut telomeres to cause DNA damage and hence trigger cell death. U6P, U6 promoter; TsgRNA, telomeric DNA-targeting sgRNA; pCMV, human Cytomegalie virus promoter; TRSE, telomerase-recognizable sticky end; MP, minimal promoter; tDNA, telomeric DNA; TF, transcription factor. Modified from [86].

## Data Availability

Not applicable.

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
