# Peer review of "CRISPR/Cas: A New Tool in the Research of Telomeres and Telomerase as Well as a Novel Form of Cancer Therapy"

_ijms, 2022, doi:10.3390/ijms23063002_

Round 1

Reviewer 1 Report

This is a review article about CRISPER/Cas as a new tool for telomere research. Actually, this is a review and there might be slightly little for readers to know something new insight into the telomere research. However, the article itself is well structured and will provide the readers of IJMS with easy access to the concept of CRISPER/Cas.

However, to move on to the next step, I strongly recommend the authors to modify their manuscript according to the guide for authors, and never change the setting of the format such as font, line space, size and so on.  

Moreover, they should delete all the unnecessary spaces in the sentences, which will increase the readability of this manuscript.

Minor points

Line 219; “NF-kB” should be correctly described.

Line 362; Figure legend for Figure 4 is required. Authors cannot state “For details, please see explanations in the text”. It is very negligent and rude to the readers.

Author Response

This is a review article about CRISPER/Cas as a new tool for telomere research. Actually, this is a review and there might be slightly little for readers to know something new insight into the telomere research. However, the article itself is well structured and will provide the readers of IJMS with easy access to the concept of CRISPER/Cas.

However, to move on to the next step, I strongly recommend the authors to modify their manuscript according to the guide for authors, and never change the setting of the format such as font, line space, size and so on.  

Moreover, they should delete all the unnecessary spaces in the sentences, which will increase the readability of this manuscript.

Response: We apologise for the partially incorrect text format which was now corrected. The text was changed from font size 10 to 9 except for headings. We removed spaces where appropriate and headings line spaces were adapted manually according to the requirements of the template.

Minor points

Line 219; “NF-kB” should be correctly described.

Response: Was corrected accordingly as it occurred during changes of general font types.

Line 362; Figure legend for Figure 4 is required. Authors cannot state “For details, please see explanations in the text”. It is very negligent and rude to the readers.

Response: We apologise for this lapsus and added a proper detailed figure legend for figure 4.

Reviewer 2 Report

The article by written by Porika et al. is an elegant review with the high educational value describing the CRISPR/Cas technique used as a tool in the telomerase activity research for cancer therapy. This manuscript is clear and well-written. The authors have described the numerous CRISPR-related methods targeting the genes relevant to telomere and telomerase and focused on their potential use in anti-cancer therapy. Their survey is complex and summarizes all the most important CRISPR/Cas techniques currently used. I wish every manuscript was written on this level. It deserves to be published in  International Journal of Molecular Sciences without any modifications except for one technical mistake described below:

Page 7, line 219 and page 8, line 261: probably the technical mistake, the “vortex” symbol used instead of the Greek letters “kappa” and “alpha”, respectively.

Author Response

The article by written by Porika et al. is an elegant review with the high educational value describing the CRISPR/Cas technique used as a tool in the telomerase activity research for cancer therapy. This manuscript is clear and well-written. The authors have described the numerous CRISPR-related methods targeting the genes relevant to telomere and telomerase and focused on their potential use in anti-cancer therapy. Their survey is complex and summarizes all the most important CRISPR/Cas techniques currently used. I wish every manuscript was written on this level. It deserves to be published in  International Journal of Molecular Sciences without any modifications except for one technical mistake described below:

Page 7, line 219 and page 8, line 261: probably the technical mistake, the “vortex” symbol used instead of the Greek letters “kappa” and “alpha”, respectively.

Response: Apologies, this was corrected accordingly as it occurred during changes of general font types.